# The incidence, maternal, fetal and neonatal consequences of single intrauterine fetal death in monochorionic twins: A prospective observational UKOSS study

R. Katie Morris[1,2]*, Fiona Mackie[3], Aurelio Tobías Garces[3], Marian Knight[4], Mark D. Kilby[2,3]

1 Institute of Applied Health Research, University of Birmingham, Edgbaston, West Midlands, United Kingdom, 2 Fetal Medicine Centre, Birmingham Women's and Children's NHS Foundation Trust, Birmingham Women's Hospital, Mindelsohn Way, Edgbaston, United Kingdom, 3 Institute of Metabolism and Systems Research, College of Medical & Dental Sciences, University of Birmingham, United Kingdom, 4 National Perinatal Epidemiology Unit, Nuffield Department of Population Health, University of Oxford, Oxford, United Kingdom

* R.K.morris@bham.ac.uk

**Data Availability Statement:** Data cannot be shared publicly because of confidentiality issues and potential identifiability of sensitive data as

## Abstract

### Objective

Report maternal, fetal and neonatal complications associated with single intrauterine fetal death (sIUFD) in monochorionic twin pregnancies.

### Design

Prospective observational study.

### Setting

UK.

### Population

81 monochorionic twin pregnancies with sIUFD after 14 weeks gestation, irrespective of cause.

### Methods

UKOSS reporters submitted data collection forms using data from hospital records.

### Main outcome measures

Aetiology of sIUFD; surviving co-twin outcomes: perinatal mortality, central nervous system (CNS) imaging, gestation and mode of delivery, neonatal outcomes; post-mortem findings; maternal outcomes.

identified within the Research Ethics Committee application/approval. Requests to access the data can be made by contacting the National Perinatal Epidemiology Unit data access committee via general@npeu.ox.ac.uk.

**Funding:** RKM, MDK, FM, MKn - no grant number, British Maternal and Fetal Medicine Society and Twins Trust bursary. https://www.bmfms.org.uk/ RKM, MDK, FM - bursary from Richard and Jack Wiseman Trust (no website) The funders had no role in study design, data collection and analysis, decision to publish, or preparation of the manuscript.

**Competing interests:** The authors have declared that no competing interests exist.

## Results

The commonest aetiology was twin-twin transfusion syndrome (38/81, 47%), "spontaneous" sIUFD (22/81, 27%) was second commonest. Death of the co-twin was common (10/70, 14%). Preterm birth (<37 weeks gestation) was the commonest adverse outcome (77%): half were spontaneous and half iatrogenic. Only 46/75 (61%) cases had antenatal CNS imaging, of which 33 cases had known results of which 7/33 (21%) had radiological findings suggestive of neurological damage. Postnatal CNS imaging revealed an additional 7 babies with CNS abnormalities, all born at <36 weeks, including all 4 babies exhibiting abnormal CNS signs. Major maternal morbidity was relatively common, with 6% requiring ITU admission, all related to infection.

## Conclusions

Monochorionic twin pregnancies with single IUD are complex and require specialist care. Further research is required regarding optimal gestation at delivery of the surviving co-twin, preterm birth prevention, and classifying the cause of death in twin pregnancies. Awareness of the importance of CNS imaging, and follow-up, needs improvement.

## Introduction

Monochorionic (MC) twin pregnancies constitute approximately 30% of all twin pregnancies and are complex due to the conjoining of the two fetal circulations by placental vascular anastomoses, predisposing the pregnancies to unique complications, including twin-twin transfusion syndrome (TTTS), selective intrauterine growth restriction (sIUGR), and single intrauterine fetal death (sIUFD) [1, 2]. Data from cohort studies and case series indicate that sIUFD complicates up to 6% of all twin pregnancies (although the true prevalence is unknown) [3]. sIUFD occurs more frequently in MC twins (7.5%) compared to dichorionic twins (3%) [4] with morbidity affecting the surviving fetus being higher in MC twins [3]. Many sIUFDs occur before 14 weeks gestation presenting at a dating ultrasound scan as a 'vanishing' twin. However sIUFDs after 14 weeks are potentially associated with serious perinatal consequences for the surviving co-twin including IUFD (i.e. after death of the first twin), preterm birth, long-term neurological comorbidity, and neonatal death [5–7]. Additionally, maternal morbidity following sIUFD has been reported with higher rates of pre-eclampsia, coagulopathy and sepsis [8, 9].

Clinical management is challenging, as controversy exists relating to optimal time of delivery, frequency of prenatal ultrasound surveillance and appropriate investigations to determine central nervous system imaging (CNS) morbidity [10, 11]. In addition, the maternal and paternal psychological effects of such a complication may be profound. We have previously published three systematic reviews investigating the outcomes of the surviving co-twin following sIUFD [5, 7, 12]; despite an additional 20 studies being able to be included in the most contemporary review and reduced heterogeneity, the same issue of small study bias persisted. Although there appeared to be an emerging consistency within the international literature supporting 'conservative management', there was little objective evidence as to: a) most reliable method of assessing fetal wellbeing, b) the use of prenatal imaging to identify CNS damage (i.e. ultrasound vs. Magnetic Resonance Imaging [MRI] or a combination), c) optimal gestation and mode of delivery.

The objectives of this study were to report on the incidence of, and maternal, fetal and neonatal complications associated with sIUFD (after 14 weeks gestation) in MC twin pregnancies in the United Kingdom. These data are important: to improve counselling of parents regarding the prognosis of the surviving co-twin, and aid management decisions.

## Methods

### Case definition

All MC twin pregnancies with a sIUFD after 14 weeks gestation, including spontaneous IUFD (i.e. no signs of TTTS, sIUGR or congenital/structural anomaly), IUFD after intervention for complications such as fetoscopic laser ablation (FLA) for TTTS, sIUGR or selective feticide (intrafetal laser, radiofrequency ablation and cord occlusion) were included. Those with "major" structural fetal anomalies (concordant and discordant) were included, as were MC monoamniotic twins. For inclusion, chorionicity had to have been confirmed in the first trimester based on the presence of the 'T' sign and absence of the 'lambda' sign, and a single placental mass [13–15]. The following were excluded: higher order multiples where fetal reduction had been performed, pregnancies with twin reversed arterial perfusion sequence. TTTS was defined according to the Quintero definition [16], and sIUGR as an inter-twin growth discordance greater than 20% in estimated fetal weights as at the time of designing the study there were no validated twin-specific growth charts.

### Outcomes

There are currently no validated core outcome sets for twin pregnancy research, thus outcomes were based on: existing literature, discussion with representatives from the Multiple Births Foundation and the Twins Trust, and limitations noted from systematic reviews [5, 7].

1. Incidence of sIUFD after 14 weeks gestation in MC twin pregnancies.

2. Complications associated with sIUFD: rates of sIUFD associated with antenatal complications (TTTS, sIUGR and congenital/structural anomalies), management of antenatal complications, rate of spontaneous sIUFDs.

3. Surviving co-twin outcomes: perinatal mortality (miscarriage defined as fetal death <24 weeks, and stillbirth fetal death >24 weeks), antenatal CNS imaging, gestation at delivery (if preterm <37 weeks, <34 weeks, <28 weeks, iatrogenic or spontaneous), role of induction, mode of delivery and reason if not normal vaginal delivery, post-mortem findings.

4. Neonatal outcomes: neonatal death defined as at least until discharge, neonatal intensive care unit admission and reason, postnatal CNS imaging, abnormal neurological signs in the neonatal period.

5. Maternal outcomes: major maternal morbidity, as reported by UKOSS studies [17].

### Data collection

Data were collected using the United Kingdom Obstetric Surveillance System (UKOSS), which prospectively captures data on severe but rare complications of pregnancy and childbirth from all obstetric-led maternity units in the UK. Each unit in the UK has a designated UKOSS clinician data collector who completes monthly returns to UKOSS. The methodology is described elsewhere [17] and is well established having been running since 2005. Additionally, the lead

of each Fetal Medicine Centre in the UK was alerted to the study. Following case notification to UKOSS by clinicians, a piloted, anonymised data collection form was sent (see www.npeu. ox.ac.uk/ukoss/current-surveillance/stwin for the form) and completed from hospital records. Reports were cross-checked for duplication based on the woman's year of birth and estimated delivery date. If forms were not returned, reminders were sent.

The surveillance period was between 1st July 2016—30th June 2017, defined as the time during which the sIUFD was diagnosed. There was a further 6 months of data collection to allow collection of delivery data. Case ascertainment was checked by comparison with MBRRACE-UK data from the same period. All deaths of twins were identified from MBRRA-CE-UK and compared with UKOSS data based on centre, month and year of reporting for case ascertainment. Where 1 potential additional case was identified, centres were contacted and asked to check whether the case met the case definition and if so to complete a UKOSS data collection form.

This study has ethical approval from the North London REC1 (REC Ref. Number: 10/ H0717/20) awarded 13[th] April 2016.

## Statistics

The incidence of sIUFD in MC twin pregnancy was planned to be calculated using ONS data as the denominator. Maternal, fetal and neonatal outcomes were described and reported as percentages, mean and SDs or medians and IQRs as appropriate. Sub-group analyses were planned to determine if TTTS or sIUGR, were associated with any characteristics, or if they were associated with a poorer outcome for the surviving co-twin. We also planned to explore prognostic indicators for mother and infant associated with sIUFD in MC twin pregnancy, e.g. gestation of IUFD of the first twin, using logistic regression. No sample size calculation was performed, but prior to commencing the study, we had estimated, based on ONS data from 2012 and existing literature, that of 11,228 twin pregnancies, 30% would be MC, and 7.5% would result in a sIUFD, thus there would be 253 cases per annum. No imputations of missing data were performed, cases with missing outcomes were not included in certain analyses as stated below. The study is funded by the British Maternal Fetal Medicine Society and Twins Trust (formerly TAMBA).

## Results

### Incidence

82 cases were reported to UKOSS in the one year study period (Fig 1). One local reporter included a trichorionic triamniotic triplet pregnancy which was excluded; 81 MC (diamniotic) twin pregnancies were included in analysis. Data were collected from 47 centres. Case ascertainment via comparison with MBRRACE-UK data identified a potential additional 94 cases which centres were asked to investigate. Twenty-eight centres replied, with most potentially eligible cases not meeting the case definition (fetal demise was outside the study period, both twins died at the same time or the twins were not MC). Only three additional cases were identified which were included in the 81 cases analysed.

The characteristics of the study population are shown in Table 1. 22/81 (27%) sIUFDs were classified as spontaneous (i.e. no signs of TTTS, sIUGR or congenital/structural anomaly), 38/ 81 (47%) pregnancies with a sIUFD were complicated by TTTS, 5/81 (6%) with sIUGR, and 12/81 (15%) with a congenital/structural anomaly. In 4/81 (5%) cases it was not clear whether there was a preceding diagnosis of a complication of MC twin pregnancy, therefore they were classified as 'unclear'. The median gestation at diagnosis of first sIUFD was 157 days, equating to 22+3 weeks (IQR 25[th]-75th: 134–186 days).

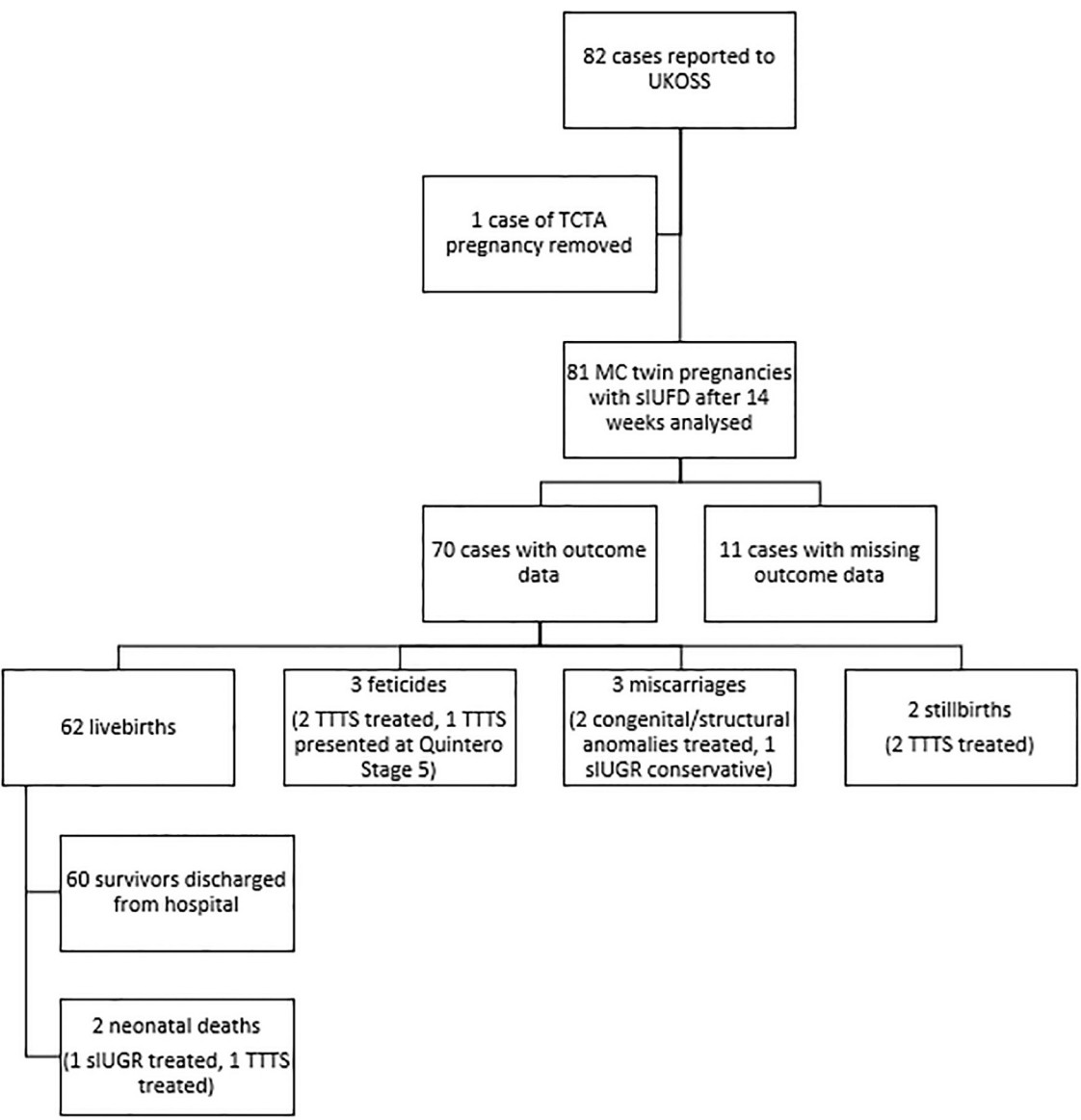

**Fig 1. Outcomes of co-twins following sIUFD in monochorionic twin pregnancies.**

### sIUFD in pregnancies that underwent antenatal interventions

Most pregnancies affected by complications related to monochorionicity had undergone pre-natal therapeutic intervention to treat a complication (43/55 [78%] pregnancies): TTTS 31/38, and sIUGR 4/5, and selective termination in 8/12 pregnancies with congenital/structural anomalies. For pregnancies with TTTS that underwent an intervention (n = 31), the median gestation of first sIUFD was 21 weeks (IQR: 6 weeks); for those that did not undergo an intervention median 20 weeks (IQR: 11 weeks). Indications for not offering antenatal intervention in 7 cases of TTTS were varied but generally related to a more complicated course of disease or late diagnosis and thus no further comparisons are made between those receiving and those not undergoing an antenatal intervention.

**Table 1. Maternal characteristics of sIUFD in monochorionic twin pregnancies.**

| Characteristic | All (n = 81) | Missing data n (%) |
|---|---|---|
| Age (years) | 29.5 (9) | 3 (3.7) |
| Ethnicity n (%) | | |
| White | 68 (88.3) | 4 (4.9) |
| Asian | 6 (7.9) | |
| Black | 2 (2.6) | |
| Other | 1 (1.3) | |
| BMI (kg/m$^2$) | 26.3 (7.0) | 8 (9.9) |
| Paid employment n (%) | | |
| Yes | 47 (69.1) | 13 (16.0) |
| No | 21 (30.9) | |
| Smoking status n (%) | | |
| Never | 53 (74.6) | 2 (2.5) |
| Stopped pre- pregnancy | 6 (8.4) | |
| Stopped during pregnancy | 6 (8.4) | |
| Current | 6 (8.4) | |
| Nulliparity n (%) | 43 (53.1) | 1 (1.2) |
| Assisted conception n (%) | 6 (7.4) | 2 (2.5) |
| Gestation at first sIUFD (days) | 157 (52) | 0 (0) |
| Presumed aetiology of sIUFD n (%) | | |
| Spontaneous | 22 (27.1) | 0 (0) |
| TTTS | 38 (46.9) | |
| sIUGR | 5 (6.1) | |
| Congenital/structural anomaly | 12 (14.8) | |
| Unclear | 4 (5) | |

Median and IQR reported unless otherwise stated.

## Spontaneous sIUFDs

For spontaneous sIUFDs (n = 22), the median gestation at first sIUFD was 24 weeks (IQR: 11 weeks) with the latest gestation a sIUFD was diagnosed at 36 weeks (Table 2).

## Surviving co-twin outcomes

Fig 1 summarises the outcomes. The outcome data for the surviving co-twin was missing in 11/81 cases (14%). 8 of the 11 cases were reported by a Fetal Medicine Centre who had treated complications, but the referring hospital where they received their ongoing maternity care did not provide outcome data, thus the data presented are the outcomes of 70 surviving co-twins. The perinatal death rate of the co-twin was 14% (10/70): there were 8 prenatal deaths and 2 neonatal deaths, equating to 62/70 livebirths (88.6%). The surviving co-twin miscarried in 3 pregnancies (2 post selective termination). There were 2 stillbirths both following FLA for TTTS in the second trimester. There were no intrapartum stillbirths. Three pregnancies with TTTS had further complications of the condition resulting in a decision for selective termination of pregnancy (e.g. abnormal CNS imaging, severe IUGR), two of which were post-intervention. There were 2 neonatal deaths due to extreme prematurity following fetal therapy related to premature rupture of membranes. All of the pregnancies with a co-twin death had a recorded complication of pregnancy (i.e. TTTS, sIUGR or congenital anomaly) and 8/10 of the deaths occurred following fetal therapy.

**Table 2. Outcomes of those ending with a livebirth following sIUFD in monochorionic twin pregnancies, grouped according to aetiology of sIUFD.**

| Outcome | All (n = 62) | Spontaneous (n = 22) | Complicated (n = 39) |
|---|---|---|---|
| Fetal sex | | | |
| Male | 26 (41.9) | 11 (50.0) | 14 (35.9) |
| Female | 36 (58.0) | 11(50.0) | 25 (64.1) |
| Birthweight surviving co-twin (g) median (IQR) | 2049 | 2215 | 1832 |
| | (819) | (588) | (1159) |
| Gestation at delivery (days) median (IQR) | 240.5 | 249 | 225.5 |
| | (39) | (26.8) | (52) |
| Delivery <37 weeks* | 48 (77.4) | 16 (77.7) | 31 (79.5) |
| Iatrogenic | 24 (50.0) | 12 (75.0) | 11 (35.3) |
| Delivery <32 weeks† | 19 (30.6) | 4 (18.2) | 15 (38.5) |
| Iatrogenic | 4 (21.1) | 2 (50.0) | 11 (73.3) |
| Induction of labour | 11 (17.7) | 3 (13.4) | 8 (20.5) |
| Mode of delivery of surviving co-twin | | | |
| NVD/Instrumental | 22 (35.5) | | 18 (46.1) |
| Breech | 2 (3.2) | 4 (18.2) | - |
| Pre-labour caesarean section | 26 (41.9) | -14 (63.6) | 11 (28.2) |
| Caesarean section after onset of labour | 12 (19.4) | 4 (18.2) | 8 (20.5) |
| NICU admission | 45 (72.6) | 19 (86.4) | 26 (66.6) |
| CNS abnormality on postnatal imaging | 8 (12.9) | 3 (13.6) | 5 (12.8) |
| Abnormal neonatal neurological signs | 4 (6.4) | 2 (9.1) | 2 (5.1) |

N (%) are reported unless otherwise stated. Excluding: the 3 pregnancies which underwent feticide of the surviving co-twin, 3 that were miscarried, 2 that were a co-twin stillbirth, and the 11 cases for which the outcome data were missing were not included. 1 pregnancy with 'unclear' aetiology was included in the 'All' column only. Data were missing for one TTTS pregnancy which delivered at 36 weeks and it was not known if delivery was spontaneous or iatrogenic.

*28/48 were <34 weeks.

†5/19 were <28 weeks.

'-' data not provided to reduce patient identifiability. 'Complicated' included pregnancies with TTTS, sIUGR and/or congenital/structural anomalies

## Antenatal CNS imaging

Of the 81 included pregnancies, local reporters were asked to confirm whether an USS or MRI had occurred specifically to look at the CNS. There were 36 women who had USS CNS scans, 31 MRI and 22 women were recorded to have had both. Abnormal CNS imaging was reported antenatally in 7 cases, in 3 cases the abnormality was confirmed on fetal MRI, in all 7 cases the imaging abnormalities were of a severity known to be associated with neurological sequelae.

## Delivery outcomes of those resulting in a livebirth

For the 70 cases with known outcomes, there were 62 livebirths (outcomes summarised in Table 2). The majority (48/62, 77%) were born before 37 weeks gestation: 23/48 due to spontaneous preterm birth, and 24/48 iatrogenic, this information was missing for 1 case. 28/62 (45%) babies were born at <34 weeks (17 spontaneous, 11 iatrogenic), and 5/62 (8%) at <28 weeks (all spontaneous). Of those who underwent induction of labour, 3/11 were induced at <37 weeks. 24/62 (39%) were born vaginally, and 38/62 (61%) by caesarean section. Seven were planned caesarean sections, 31/38 were emergency caesarean sections, the commonest indication for emergency caesarean section was fetal distress (13/31).

## Neonatal outcomes and postnatal CNS imaging

Most neonates (45/62 babies; 73%) were admitted to the neonatal/special care baby unit, the commonest reason for admission was prematurity (29/45 babies; 64%). The median length of stay in the neonatal/special care baby unit was 11.5 days (IQR: 5–34 days). Only 9/62 (15%) babies underwent postnatal CNS imaging: imaging in 8/9 babies was abnormal, and normal in 1 baby. It was not known if postnatal CNS imaging occurred later in 22/62 babies (35%), the remaining 31/62 (50%) did not have postnatal CNS imaging. Of the 8 babies with abnormal postnatal CNS imaging, 1 baby's ultrasound was repeated and was subsequently was reported as "normal"; the remaining 7 were all were delivered at <36 weeks (median: 33+0 weeks, IQR: 30+3–34+2 weeks) with a median birthweight of 1980g (IQR:1610-2085g). Of these, 3/7 had undergone antenatal CNS imaging although the results were unknown by the reporter, the other 4 had not undergone antenatal CNS imaging. There was no association with aetiology of the initial sIUFD: 3 spontaneous at 32–34 weeks, 3 TTTS treated, 1 congenital/structural anomaly. 4/62 liveborn babies had evidence of abnormal CNS signs, all of which had abnormal postnatal CNS imaging. Only 12/60 (20%) babies discharged home were reported to have follow-up planned, 45/60 (75%) were reported to have no follow-up, these data were unknown in 3/60.

## Post-mortem examination

Post-mortem examination was performed on 17/70 (24%) pregnancies for whom there were known outcomes. 4 cases were double IUFD (both twins died). The post-mortem was able to provide a cause of death in only 3/17 (18%) cases: 'acute twin to twin transfusion', chronic TTTS, and extreme prematurity. Post-mortem examination was inconclusive in 4 cases, the fetus was unable to be examined in 2 cases due to severe maceration, and data were missing in the 8 remaining cases. The 2 neonatal deaths did not undergo post-mortem.

## Maternal outcomes

4/70 women (6%) with known outcomes were admitted to intensive care [ITU] (critical care level 3). Of these women, 3 women were admitted with sepsis (with a background of chorioamnionitis). Two of these had previously undergone invasive, prenatal intervention for MC twin complications earlier in pregnancy and were complicated by premature rupture of membranes. There was more than 6 weeks between the procedure and ITU admission. One woman was admitted for suspected "mirror syndrome" (severe pre-eclampsia in association with a hydropic fetus). The maximum length of stay in ITU was 2 days (range 1–2 days). No maternal deaths or other major maternal morbidities were reported.

# Discussion

## Main findings

The commonest associated aetiology with sIUFD in MC twin pregnancies was TTTS and the majority had undergone prenatal therapy prior to sIUFD (as opposed to being conservatively managed). The second most common aetiology was "spontaneous" with no obvious underlying aetiology (i.e. no signs of TTTS, sIUGR or congenital/structural anomaly), followed by congenital/structural anomaly, and sIUGR being the least common. Death of the co-twin following sIUFD was common, complicating 1 in 7 pregnancies. Less than two thirds of cases had investigation by antenatal CNS imaging to identify ischaemic brain injury. Of the cases whose imaging results were known to the UKOSS reporter, 1 in 5 had radiological findings suggestive of neurological morbidity. Postnatal CNS imaging revealed a further 7 babies with

brain abnormalities (mainly on cranial USS), all of which were born at <36 weeks, and 4 of whom also had abnormal CNS signs. Preterm birth was the commonest adverse outcome, with three quarters of twins born at less than 37 weeks gestation, and an equal split between spontaneous preterm birth, and those delivered iatrogenically. Major maternal morbidity was not uncommon, with 6% requiring ITU admission, compared to 0.22% of the general obstetric population reported in the 2015/2016 National Maternity and Perinatal Audit [18]. Interestingly, all maternal admissions were related to sepsis, although not all women had undergone prenatal 'invasive' interventions for therapy.

## Strengths and limitations

To the authors' knowledge, this is the first UK-wide study of sIUFD in MC twin pregnancies. A major strength of this study is that data were collected from UK centres, thus the results represent current UK-wide practice. UKOSS is a validated mechanism of collecting data on relatively rare conditions that is UK wide. Data collection relies on a system of designated clinician data collectors. In this study there were 8 cases for whom outcome data was missing following fetal therapy at a tertiary centre. This is in contrast to the only other UKOSS study relating to a fetal condition, gastroschisis, where there were no missing outcome data. This difference might reflect the complexity of complicated MC twins requiring management at multiple care providers (supraregional, regional and local), and that within the gastroschisis study outcome data was also collected via paediatric surgeons and via cross-checking with congenital anomaly registers. One of the difficulties of data collection in complex conditions is a lack of standard pathways of care and definitions of outcomes (e.g. ultrasound findings). This is especially true for the complex conditions complicating MC pregnancies, where each fetus maybe affected differently. Since data collection for this study there have been initiatives to guide the management of these complex pregnancies, and define outcomes [19, 20] and within the UK specialist fetal medicine commissioning will improve pathways of care.

The number of cases notified was considerably lower than our pre-study estimate. However, MBRRACE-UK data (national surveillance of perinatal deaths) in 2016, noted perinatal mortality rates amongst twins as the lowest ever reported [21, 22] and thus we employed triangulation of data with MBRACCE-UK to check case ascertainment. Using ONS data [23] and MBRACCE data [21], and newer evidence to suggest that 20% of twins are monochorionic [24, 25] (as opposed to the 30% of our original estimate), we calculated incidence of single twin demise across the 2 years of our study. This gives very similar figures of an estimated 75 cases in 2016 and 84 in 2017 with incidence of 37.6–38.7 per 1000 monochorionic maternities. A further strength of this study was the collection of data relating to interventions, management and outcomes for both mother and neonate. It is not possible to collect data for the baby via the UKOSS system past the point of discharge from hospital after birth. Further studies and research in this area should include data related to longer-term outcomes for the baby [7].

## Interpretation

The lower than expected number of cases reflects the decrease seen in twin pregnancy perinatal mortality reported by MBRRACE-UK in 2013–2016 [2]. It is thought this decrease may be a consequence of improved care for MC twin pregnancies, particularly in recognising and treating complications unique to MC twins, and development of new treatment techniques including the Solomon Technique for FLA [26]. This is also linked to updated national and international guidance [13, 15, 19], which the Twins Trust has demonstrated that implementation leads to lower adverse outcome rates [27]. However, the latest MBRRACE-UK report published in October 2019 reported an increase in the stillbirth and neonatal death rates in twin

pregnancies in 2017 [22]. It is important to consider data over a longer period alongside the use of three-year rolling averages to better reflect trends in perinatal mortality [2].

MC twin pregnancies complicated by sIUFD have a high risk of subsequent co-twin neurologic morbidity. From these data it appears that prenatal and indeed postnatal screening for abnormalities of the central nervous system in survivors is not routine in the UK. There is a need to strengthen professional guidance and practice amongst both obstetricians and neonatal paediatricians [19].

This study strengthens the argument made in our 2019 systematic review and meta-analysis for the need for a purposely designed prospective study, ideally with antenatal and postnatal imaging and long-term follow-up [7].

There is no specific guidance regarding when and how co-twins after sIUFD should be delivered. Of note in this study there was a high rate of preterm birth (both spontaneous and iatrogenic) and a high rate of emergency caesarean sections. The majority of co-twin survivors were admitted to the neonatal unit with prematurity the commonest indication. Despite the high risk nature of the pregnancy and requirement for admission to NNU, very few babies had planned follow-up despite an association with adverse long-term outcomes [10].

The uptake of post-mortem examination was low (25.4%) for the initial sIUFD but increased to 40% if the co-twin died as well. This is below the 75% recommended uptake by the RCOG [28], and may reflect that parents accept that MC twin pregnancies are higher risk, and even if a cause was not apparent antenatally, the findings of the post-mortem are unlikely to affect a subsequent pregnancy as it may be linked to monochorionicity. Despite post-mortem being considered the most useful investigation for parents to find out why their baby died [29], in 6/9 post-mortems in which the UKOSS reporter knew the findings, the post-mortem was inconclusive. This highlights another area of future research, as there is not currently a specific classification system for cause of death in MC twins, which is often different to the cause of death in singletons, and is the classification system which pathologists have to currently use. Since performing this study, we have proposed a new classification system of causes of death in twin pregnancies (CoDiT) which requires further validation [30]. These findings also raise the consideration of whether specialist perinatal pathologists are needed for MC twin pregnancy post-mortems, and whether injection studies should be performed in all MC twin pregnancies to aid determining cause of death.

## Conclusion

MC twin pregnancies with sIUFD remain complex and high-risk. They should not be treated as low-risk singleton pregnancies. Women at risk of developing sepsis (prenatal therapy) should be closely monitored. It is not clear what gestation to deliver the surviving co-twin; this should be investigated with a randomised controlled trial but given the relative rarity of sIUFD this is unlikely to occur. Preterm birth was the commonest adverse outcome, and more research is required in this area. Awareness of the importance of CNS imaging, and follow-up, needs to be increased, and may be aided by better communication between specialities and hospitals.

## Acknowledgments

We thank all the clinicians who contributed to the UKOSS data collection, the study would not have been possible without them.

## Author Contributions

**Conceptualization:** R. Katie Morris, Marian Knight, Mark D. Kilby.

**Data curation:** R. Katie Morris, Marian Knight, Mark D. Kilby.

**Formal analysis:** R. Katie Morris, Fiona Mackie, Aurelio Tobías Garces, Marian Knight, Mark D. Kilby.

**Funding acquisition:** R. Katie Morris, Marian Knight, Mark D. Kilby.

**Investigation:** R. Katie Morris, Marian Knight, Mark D. Kilby.

**Methodology:** R. Katie Morris, Marian Knight, Mark D. Kilby.

**Project administration:** R. Katie Morris, Fiona Mackie, Marian Knight, Mark D. Kilby.

**Resources:** R. Katie Morris, Marian Knight, Mark D. Kilby.

**Software:** Marian Knight.

**Supervision:** R. Katie Morris, Mark D. Kilby.

**Validation:** R. Katie Morris, Fiona Mackie, Aurelio Tobías Garces, Marian Knight, Mark D. Kilby.

**Writing – original draft:** R. Katie Morris, Fiona Mackie, Marian Knight, Mark D. Kilby.

**Writing – review & editing:** R. Katie Morris, Fiona Mackie, Aurelio Tobías Garces, Marian Knight, Mark D. Kilby.

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
