## [Decision Letter · Decision Letter 0]

19 Aug 2020

PONE-D-20-19757

The incidence, maternal, fetal and neonatal consequences of single intrauterine fetal death in monochorionic twins: a prospective observational UKOSS study

PLOS ONE

Dear Dr. Morris,

Thank you for submitting your manuscript to PLOS ONE. After careful consideration, we feel that it has merit but does not fully meet PLOS ONE’s publication criteria as it currently stands. Therefore, we invite you to submit a revised version of the manuscript that addresses the points raised during the review process.

thank you for submitting this interesting study please response to the reviewers comments in addition please add the interval time between sIUFD and delivery as this may be pertinent when looking at outcomes of maternal and fetal wellbeing

We look forward to receiving your revised manuscript.

Kind regards,

Andrew Sharp, PhD

Academic Editor

PLOS ONE

Journal Requirements:

3. Your ethics statement must appear in the Methods section of your manuscript.

If your ethics statement is written in any section besides the Methods, please move it to the Methods section and delete it from any other section.

Please also ensure that your ethics statement is included in your manuscript, as the ethics section of your online submission will not be published alongside your manuscript.

Reviewers' comments:

Reviewer's Responses to Questions

**Comments to the Author**

1. Is the manuscript technically sound, and do the data support the conclusions?

Reviewer #1: Yes

2. Has the statistical analysis been performed appropriately and rigorously? 

Reviewer #1: Yes

3. Have the authors made all data underlying the findings in their manuscript fully available?

Reviewer #1: Yes

4. Is the manuscript presented in an intelligible fashion and written in standard English?

Reviewer #1: Yes

5. Review Comments to the Author

Reviewer #1: This prospective UK-based study is novel, the topic is of high clinical importance for subspecialists caring for twins and the figures described are useful for counselling. The manuscript is well written and conclusions are clear and logical.

The number of cases identified was significantly lower than the estimated 253, but the authors have addressed this well in the discussion and it does not appear that a significant number of cases are likely to be missed.

A few questions (I appreciate this data may not be available but I think would be of interest to readers if it could be included):

Of those with TTTS, what were the details for mean gestation etc? Were there any noticeable differences in the deaths between those that did and did not have intervention? Was there anything to add about gestations at which fetal therapy was performed and Staging of TTS at time of intervention.

For the sepsis cases what was the timeframe relationship to the IUFD and did this result in second twin demise?

What were the causes of the two NND in co-twins (from certificates)?

Is there anything to explain why only 36 have brain imaging antenatally? Was TOP offered for the 7 with findings that were abnormal antenatally? If so, how many accepted TOP? Similarly, is there any indication of why only 9/62 in NNU have brain imaging? I think the lack of provision of CNS imaging and CNS follow-up for surviving twins is the most striking finding of the study and it would be good to add these details if they are available and also to discuss how the authors would propose to address this variation in provision across the UK.

6. PLOS authors have the option to publish the peer review history of their article (what does this mean?). If published, this will include your full peer review and any attached files.

Reviewer #1: No

---

## [Author Response · Author response to Decision Letter 0]

2 Sep 2020

Detailed responses in Response to Reviewers document. Updated data sharing statement provided.

---

## [Editor Report · Decision Letter 1]

7 Sep 2020

The incidence, maternal, fetal and neonatal consequences of single intrauterine fetal death in monochorionic twins: a prospective observational UKOSS study

PONE-D-20-19757R1

Dear Dr. Katie Morris,

We’re pleased to inform you that your manuscript has been judged scientifically suitable for publication and will be formally accepted for publication once it meets all outstanding technical requirements.

Kind regards,

Andrew Sharp, PhD

Academic Editor

PLOS ONE
---

## [Editor Report · Acceptance letter]

11 Sep 2020

PONE-D-20-19757R1

The incidence, maternal, fetal and neonatal consequences of single intrauterine fetal death in monochorionic twins: a prospective observational UKOSS study

Dear Dr. Morris:

I'm pleased to inform you that your manuscript has been deemed suitable for publication in PLOS ONE. Congratulations! Your manuscript is now with our production department.

Kind regards,

on behalf of

Dr. Andrew Sharp 

Academic Editor

PLOS ONE